# Regulatory Effects of Curcumin on Platelets: An Update and Future Directions

**DOI:** 10.3390/biomedicines10123180

**Published:** 2022-12-08

**Authors:** Yaseen Hussain, Fazlullah Khan, Khalaf F. Alsharif, Khalid J. Alzahrani, Luciano Saso, Haroon Khan

**Affiliations:** 1Lab of Controlled Release and Drug Delivery System, College of Pharmaceutical Sciences, Soochow University, Suzhou 215000, China; 2Department of Pharmacy, University of Malakand, Chakdara 18800, Pakistan; 3Faculty of Pharmacy, Capital University of Science & Technology, Islamabad 44000, Pakistan; 4Department of Clinical Laboratory, College of Applied Medical Science, Taif University, P.O. Box 11099, Taif 21944, Saudi Arabia; 5Department of Physiology and Pharmacology “Vittorio Erspamer”, Sapienza University, 00185 Rome, Italy; 6Department of Pharmacy, Abdul Wali Khan University Mardan, Mardan 23200, Pakistan

**Keywords:** curcumin, platelets, antioxidants, anti-inflammatory, therapeutic potential

## Abstract

The rhizomatous plant turmeric, which is frequently used as a spice and coloring ingredient, yields curcumin, a bioactive compound. Curcumin inhibits platelet activation and aggregation and improves platelet count. Platelets dysfunction results in several disorders, including inflammation, atherothrombosis, and thromboembolism. Several studies have proved the beneficial role of curcumin on platelets and hence proved it is an important candidate for the treatment of the aforementioned diseases. Moreover, curcumin is also frequently employed as an anti-inflammatory agent in conventional medicine. In arthritic patients, it has been shown to reduce the generation of pro-inflammatory eicosanoids and to reduce edema, morning stiffness, and other symptoms. Curcumin taken orally also reduced rats’ acute inflammation brought on by carrageenan. Curcumin has also been proven to prevent atherosclerosis and platelet aggregation, as well as to reduce angiogenesis in adipose tissue. In the cerebral microcirculation, curcumin significantly lowered platelet and leukocyte adhesion. It largely modulated the endothelium to reduce platelet adhesion. Additionally, P-selectin expression and mice survival after cecal ligation and puncture were improved by curcumin, which also altered platelet and leukocyte adhesion and blood–brain barrier dysfunction. Through regulating many processes involved in platelet aggregation, curcuminoids collectively demonstrated detectable antiplatelet activity. Curcuminoids may therefore be able to prevent disorders linked to platelet activation as possible therapeutic agents. This review article proposes to highlight and discuss the regulatory effects of curcumin on platelets.

## 1. Introduction

The chemical name of curcumin, a phenolic molecule, is (1*E*,6*E*)-1,7-bis (4-hydroxy-3-methoxyphenyl)-1,6-heptadiene-3,5-dione [1]. It is also known as diferuloylmethane. It has been classified as both a PAINS (pan assay interference compounds) and an IMPS (invalid metabolic panaceas) candidate [2]. The rhizome and roots of the turmeric plant contain significant levels of this biologically active chemical (*Curcuma longa*) [3]. Demethoxycurcumin and bis-demethoxycurcumin are two more curcuminoids found in this plant in addition to curcumin (Figure 1). It has been found that 2–4 percent of the dry turmeric root powder is made up of curcuminoids. Curcumin is a yellow substance that has a variety of uses, including herbal dietary supplements and culinary coloring and flavoring agents in various preparations [4].

Curcumin was first isolated in an impure form in 1815. Curcumin demonstrates keto-enol tautomerism in its position as a bis-α, β-unsaturated β-diketone. While curcumin’s keto form is more common in acidic and naturally occurring pH, the enol form is more common in alkaline solutions [5,6]. It is important to note that Curcumin is employed in the food, pharmaceutical, and textile industries. According to evidence dating back 2500 years, curcumin has long been used to cure different diseases in traditional Indian medicine and throughout Asia. Curcumin has been considered a potent remedy for various ailments, including rheumatism, dyspepsia, and irritable bowel syndrome as well as infection treatments for eye and skin conditions [7,8].

Since 1937, when the first study on curcumin’s medicinal effects was published, much effort has gone into illuminating the compound’s unique pharmacological properties [9]. According to reports, curcumin has beneficial biological and therapeutic properties including antioxidant, hepatoprotective, cardioprotective, neuroprotective, anti-microbial, anti-tumor and improvement of dyslipidemia and ischemia-reperfusion [10]. In oil-based solutions, curcumin is very soluble. Curcumin is soluble at alkaline pH despite being insoluble in water at acidic and neutral pH levels. As was already established, although having outstanding therapeutic effects, curcumin’s effectiveness has been significantly constrained by several factors [11]. Its limited solubility in aqueous environments and quick conversion into an inactive metabolite are the biggest obstacles. Due to this, numerous studies have focused on improving the solubility of curcumin or encapsulating it to create targeted drug delivery systems for biological uses [12].

Platelets are non-nucleated cells of the blood generated continuously by the human body (10^11^ cells daily) from megakaryocytes via differentiation, maturation, and fragmentation [13]. Usually, their normal life span ranges from 8 to 10 days in the blood [14].

Thrombopoiesis (the formation of platelets) occurs in bone marrow that involve the formation of megakaryocytes (50–100 μm in diameter) from hematopoietic stem cells [15]. The mature megakaryocytes shed long-branching cytoplasmic protrusions, called proplatelets [16]. When the platelet count is reduced, thrombopoietin is secreted by the liver that activates thrombopoietin receptor of megakaryocytes resulting in the proplatelets generation [17]. Situations requiring thrombopoiesis, for instance during inflammation, IL-6 increases the thrombopoietin level, thereby increasing the formation of proplatelets [18]. The generation of proplatelets involves the redistribution of granules, vesicular structures, and organelles from the cell body of megakaryocytes into proplatelets. The proplatelets subsequently give rise to platelets. It has been found that proplatelet formation is inhibited by collagen I (via glycoprotein VI (GPVI) and integrin α2β1) and stimulated by fibrinogen, type IV collagen, and fibronectin [19].

Platelets contain different cell organelles, including ribosomes, mitochondria, microtubules, glycogen granules, occasional Golgi elements and enzyme systems for aerobic as well as anaerobic respiration [20,21]. In addition, one of the important organelles is their granules, which are of three types known as α-granules, δ-granules (dense granules), and lysosomes. The α-granules are the largest and most abundant (50–80 α-granules/platelet) [22], heterogenous in content and function that contain adhesive proteins including von Willebrand Factor (vWF), platelet factor 4 (PF4), fibronectin, fibrinogen, platelet derived growth factor (pdGF), tetraspanins (e.g., CD 9) and immunoglobulin family receptors (e.g., PECAM, GPVI) [23]. The δ-granules are smaller, less in number (3–8/platelet), electron-dense, containing non-protein molecules involved in platelet activation and vasoconstriction such as calcium, serotonin, histamine, ATP, and ADP [24,25], while the lysosomes are scanty, membrane-bound vesicles containing lysosomal enzymes (e.g., hexosaminidase and cathepsins). These act in the digestion of cytosolic components and extracellular functions, such as degradation of extracellular matrix, cleavage of receptors and fibrinolysis [26]. Recently, a novel, electron-dense, secretory granule designated as T granule has been reported [27], which behaves similar to the Toll-like receptors (TLR). Defects or deficiency in platelet granules have been linked to several bleeding disorders, such as Hermansky-Pudlak Syndrome (dense granule deficiency) and Grey Platelet Syndrome (α-granule deficiency) [28].

Platelets form a platelet plug, thus blocking bleeding during vascular injury [29]. It plays an important role in infection, tumor growth, vascular repair and constriction, inflammation, and tissue homeostasis [30,31,32,33]. Platelets help to regulate the tone of blood vessels, re-formation of neointima after injury to wall of the vessels, and pathological processes such as atherosclerosis, cancer metastasis, and thrombosis [34]. Oxidative stress has been implicated in the modulation of platelet function resulting in platelet aggregation [35]. The elevated ROS concentration reduce the amount of NO., an antiplatelet and vasorelaxant agent, forming a cytotoxic substance, peroxynitrite [36]. Oxidative stress and the subsequent chronic inflammation are involved in several diseases including cardiovascular disorders (CVDs), and cancer.

It has been observed that curcumin prevent platelet activation and aggregation, most likely through inhibition of lipoxygenase and cyclo-oxygenase pathway (thus blocking thromboxane B2 and 12-HETE production) [37], interfering with cytosolic Ca^2+^ which is necessary for platelet activation and aggregation [38]. Other mechanisms of curcumin mediated platelets inhibition involve reduction of mitochondrial membrane potential, microparticle formation, and increased annexin-V binding [39]. According to some reports, curcumin prevents its formation of fibrinogen and thus, lowers plasma levels of fibrinogen. Thus, curcumin administration to HFD-rats resulted in lower fibrinogen and platelet counts, indicating a physiological response to preventing thrombosis and lowering CVD risk [40]. In the present review, we aimed to highlight the regulatory effects of curcumin on platelets.

## 2. Functions of Platelets

### 2.1. Hemostasis

Hemostasis is the ability to prevent bleeding from an injured blood vessel. It may be primary, secondary, or fibrinolysis [41]. Primary hemostasis involves stopping of blood loss via formation of platelets plug [27]. Secondary hemostasis involves the deposition of insoluble fibrin produced through coagulation cascade. Fibrinolysis is the breakdown of fibrin clot through several enzymes that occur during wound healing [41]. In normal healthy vessels, the endothelium is intact and offers a non-adhesive surface to platelets, while in case of damage, the platelets adhere to the extracellular matrix of the exposed endothelial surface and thus form a platelet plug, which is a three-step process including platelet adhesion, platelet activation and secretion, and platelet aggregation [42].

### 2.2. Platelet’s Role beyond Hemostasis

a.Inflammation and immunity

Platelets play an important role in immunity and inflammation by interacting with immune cells [43]. The α- and dense granules contain proinflammatory mediators including growth factors, TGF-β, P-selectin, and CD40L. These cause stimulation of other cells to secrete proinflammatory chemokines and cytokines such as TNFα, IL-8, IL-1β, and CCL2 [44]. Platelets contain several Toll-like receptors (TLR) ranging from TLR1-9 that identify molecular motifs known as pathogen-associated molecular patterns (PAMPs) and thus stimulate immune system [45]. Bacteria, via TLR2 stimulation, initiate a pro-inflammatory response through the PI3K signaling pathway [22]. Likewise, TLR9 stimulation is linked to oxidative stress and thrombosis [27].

Recently, a link between immune system and platelets has been found that involve the interaction of platelets with complement system [46,47,48]. Platelets bind C3b, a key complement component, via P-selectin and activates the formation of membrane attack complex and anaphylatoxin C5a, that is responsible the lysis of pathogen cells [49].

Platelets have the intrinsic ability to store and release significant quantities of chemokines and cytokines involved in inflammation [50]. Platelets are the first cells to reach at the infection site in blood vessels. They are also indispensable players in infection and immune response to viral and bacterial infections [51]. Deficiency of platelets (thrombocytopenia) occur in sepsis, and it has been proved that platelets exhibit important role in multiorgan failure and sepsis [52].

b.Cancer

Platelets play an important role in the pathogenesis of metastasis [53]. It has been observed in the in vitro studies that platelets adhere to metastatic cells, thus forming a “cloak” around the circulating tumor cells and hence, acting as a shield for immune clearance. This has resulted in epithelia-mesenchymal transition, pro-angiogenic, and pro-survival effects in cancer cells [54]. Moreover, platelets have been found as the culprits behind the enhanced tumor growth by secretion of PDGF and VEGF [55].

c.Wound healing

It has been found that platelets are relevant mediators of tissue regeneration and wound healing. It involves the release of growth factors and several other mediators of repair and regeneration, including cytokines, fibronectin, vitronectin, and sphingosine 1-phosphate. The different steps involved in wound healing includes hemostasis, inflammation, proliferation, and remodeling/maturation. Platelets are key players in these steps [56]. As the tissue is injured, platelets quickly form fibrin clot that halts bleeding [57]. Platelets and neutrophils help to resolve inflammation by secreting several pro-resolving mediators and polarizing macrophages towards a repair phenotype [58]. Furthermore, platelets play important role in the proliferative phase of repair by releasing angiogenic and growth factors. Angiogenesis: an important step to cope with increased metabolic needs of the healing tissues; is induced by VEGF, Fibroblast Growth Factor (FGF), Hepatocyte Growth Factor (HGF); platelets activate the recruitment of CD34+ bone marrow derived endothelial progenitors [59]. Moreover, platelets release PDGF and TGF-b that act on fibroblasts so that the initial provisional fibrin scaffold is replaced with a granulation tissue rich in immature collagens, proteoglycans, and fibronectin [60]. Finally, platelets help to remodel the extracellular matrix by releasing matrix metalloproteinase and hydrolases. Platelets-rich plasma (PRP) gels are available that is applied in clinics for healing diabetic ulcers and skin wounds [61].

d.Infection

Platelets play an important role in infection ranging from directly killing bacteria to increasing the differentiation of immune cells [32,62]. Platelets granules contain proteins, such as thrombocidin 1 and 2 which kill a broad range of bacteria by direct microbicidal effect [63,64]. It has been proved that platelets detect lipopolysaccharides of bacteria by toll-like receptor 4 and activate the neutrophil extracellular traps (NET) formation in neutrophils [65]. NET is composed of neutrophilic proteins, histones, and DNA that traps and eliminates fungi and bacteria [66,67]. Moreover, platelets contain β1-defensins that stimulate the production of NET and stop bacterial growth [68]. P-selectin in platelets stimulates platelets-dependent NET formation [69]. Likewise, platelets contain a ligand for leucocyte triggering receptor expressed on myeloid cells 1 (TREM-1). Bacterial structures upregulate these receptors that enhance the secretion of IL-8 by the neutrophils and enhance TREM-1-induced respiratory burst [70]. Furthermore, T-lymphocytes are the main players in modulation of immune response by platelets. In an acute viral hepatitis model, platelets have been found to trigger cytotoxic T-cells response, resulting in hepatic injury [71]. Moreover, the interaction of cytotoxic T-cells and platelets depend on platelets CD154 [72,73]. Similarly, in chronic viral hepatitis model, the serotonin derived from platelets exacerbated hepatocytes damage by reducing cytotoxic T-cells recruitment and sinusoidal blood flow [74]. It is noteworthy that serotonin stimulate T-cells through 5-HT receptors [75]. In addition, a chemokine, Regulated And Normal T cell Expressed and Secreted (RANTES), released by platelets has been found to play pivotal role in cytotoxic T-cells function in viral infections [76].

## 3. Pharmacological Effects of Curcumin

Many in vivo and in vitro model systems were used in past decades to identify the pharmacological effects of curcumin. However, its poor pharmacokinetics has led to its limited use in humans following clinical applications. Curcumin exhibits many versatile pharmacological effects that are schematically shown in Figure 2.

Curcumin exhibits promising antioxidant activity. It has been shown that curcumin mimics the level of oxidative stress markers systemically thus leads to the modulation of enzymes responsible for free radical neutralization such as catalase, superoxide dismutase, glutathione etc. [77] in addition, curcumin also exhibit free radical scavenging activity and scavenge nitrogen and oxygen reactive species [78]. Results of a recent meta-analysis study showed that curcumin significantly decreased malondialdehyde concentration in participant subjects and increased the total antioxidant capacity showing the antioxidant potential of pure curcumin [79]. In another study, curcumin loaded Zein/carboxymethyl dextrin nanoparticles were evaluated and results showed impressive antioxidant potential of curcumin from nanotechnology platform [80]. Results from reducing power assay and DPPH radical scavenging analysis showed a high in vitro antioxidant activity for curcumin loaded whey protein micro gels [81]. Curcumin is a well-known anti-inflammatory agent. In this regard, using ovalbumin-induced allergic asthma mouse model, curcumin was administered to BALB/c mice in a dose of 20 mg and 100 mg/kg dose. From results, it was concluded that curcumin showed an anti-inflammatory effect through suppression of pro inflammatory cytokines and elevation in the expression levels of aquaporin [82]. Glucan particles extracted from yeast were used for loading curcumin and were delivered in vitro to evaluate its anti-inflammatory potential. The secretion of pro-inflammatory cytokines, TNF-α and IL-1β, showed the effective anti-inflammatory response of the delivered curcumin [83].

Defensive mechanisms of the immune system combat infections [84]. The activity of the immune system is modulated by immunomodulators that in turn reduce inflammation through normalization of the immune system [85]. The normal function of the immune system is affected by many flavonoids posing their pharmacological action [86]. The expression of proinflammatory cytokines and chemokines was down regulated by curcumin via NF-kB inactivation [87]. At low doses curcumin showed modulation of immune system and showed its pharmacological effect in various ailments, i.e., cancer, heart diseases, asthma, and diabetes [88]. In an experimental multiple sclerosis animal model, curcumin resulted in decreased production of IL-2 as well as STAT4 activation and showed an immunomodulatory effect [89]. In addition, data from pre-clinical and clinical trials have shown the curcumin immunomodulatory actions focusing mediators and immune cells involved in immune responses [90,91]. Curcumin has a versatile immunomodulatory action and affects different immune cells, showing its future use in immune diseases therapy.

The pathological process of neuronal killing is called excitotoxicity [92]. Excessive glutamate induces calcium influx and neuronal injury [93]. Eventually, excitotoxicity associated neurodegeneration and depressive disorders are triggered [94]. In such major depressive disorders, curcumin has shown antidepressant effect [95]. Curcumin also decreases calcium influx along with inhibition of A-kinase anchoring protein 79 translocation from cytomembrane to cytoplasm [96]. Another study showed that curcumin significantly elevated the expression of brain-derived neurotrophic factor along with cell viability [97]. Similarly, curcumin suppressed the neuroprotective effect and enhanced the expression of brain-derived neurotrophic factor with retardation of the TrkB signaling pathway [98]. Programmed cell death is achieved through a regulated mechanism known as apoptosis [99]. Curcumin exhibited anticancer and chemo preventive potential through cell cycle arrest and eventually leads to apoptosis using various intrinsic and extrinsic pathways [100]. In neuronal cells, β-amyloid results in the induction of apoptosis and curcumin significantly showed antiapoptotic activity in such cells [101].

## 4. Effects of Curcumin on Platelets

Curcumin exerts antiplatelet activity in several pathological conditions, including inflammatory diseases, atherthrombosis, and thromboembolism, thus playing a pivotal role in cardiovascular diseases (CVDs). Effects of curcumin on platelets is shown in Table 1.

### 4.1. Coagulation and Angiogenesis

Platelets can promote coagulation by exposure of phosphatidylserine (phospholipid) [102] and also promote critical process including angiogenesis and metastasis in caner [103]. Inflammation activates the procoagulant molecules and alters the coagulation system. Platelets secrete IL-1β consequently secreting cytokines dependent on IL-1β such as IL-4, 6 and 8, which are among the main pro-inflammatory cytokines of inflammation [104]. Moreover, thrombin mediates platelet activation via P-selectin expression [105]. In rats, intraperitoneal administration of curcumin (60 mg/kg) reduced mortality in lipopolysaccharide (LPS)-induced intravascular coagulation by decreasing TNF-α level. This research study demonstrated the beneficial potential of curcumin in coagulopathy induced by infections [106]. Furthermore, it has been shown that curcumin inhibits platelets aggregation induced by collagen, adrenaline, and arachidonic acid. It also causes suppression of thromboxane B_2_ production and elevate 12-lipoxygenase (LOX) enzyme. Curcumin inhibits the expression of platelet/endothelial cell adhesion molecule 1, netrin G1, delta-like 1, and plasma cell endoplasmic reticulum protein 1, which cause cell adhesion and migration [107].

**Table 1 biomedicines-10-03180-t001:** Effects of curcumin on platelets.

S.No.	Parameter	Effect/Mechanism	References
1	Coagulation	Inhibit coagulation, ↓ TNF	[106]
2	Platelet aggregation	Inhibit platelet aggregation and platetlet plug formation, ↓ cell adhesion molecule 1, netrin G1, delta-like 1, and plasma cell endoplasmic reticulum protein-1	[107]
3	Platelets activation	Inhibit activation of platelets to form thrombosis/embolism, ↓ P-selectin, E-selectin, and GP IIb/IIIa	[108]
4	Autophagy	inhibition of PKB, and activation of AMP kinase	[109]
5	Antioxidant effect	↑ antioxidant enzymes, ↓ oxidative stress parameters, ↑ platelet factor-3-like activity	[110,111]
6	Platelet count	↑ platelets level	[112]
7	PDGF	Ameliorated lung fibrosis, liver fibrosis, and cirrhosis, Inhibit PDGF	[113]
8	Platelet aggregation and hyperlipidemia	↓ cholesterol, ↑ antioxidant activity	[114]
9	Atherosclerosis	Thromboxane inhibition, ↑ prostacyclin activity	[115]
10	Arachidonic acid-mediated platelet aggregation	Inhibition of TXA2 and mobilization of intracellular Ca^2+^	[116]

### 4.2. Activation of Platelets

Coagulation cascade and platelets activity are linked with each other. Glycoprotein IIb (GPIIb)/IIIa receptor activation play key role in the aggregation of platelets [117]. The procoagulant platelet response is also facilitated by the adhesive complexes glycoprotein Ib-V-IX and integrin αIIbβ3 [118]. The key pathway for platelet activation is via activation of receptor GP IIb/IIIa that cause cross-linking of von Willebrand factor or fibrinogen between receptors leading to platelets aggregation [117]. Administration of curcumin inhibits platelets adhesion and elevation of GP IIb/IIIa mediated platelet activation that is associated with decreased expression of P-selectin, E-selectin, and GP IIb/IIIa on platelets as shown in Figure 3 [108]. Curcumin also inhibits platelet activation by interfering with spleen tyrosine kinase and subsequent activation of phospholipase C gamma [119]. Curcumin has shown anticoagulant activity in vitro [120]. Moreover, in-vivo study exhibited that curcumin inhibited platelet aggregation in monkeys. This implies that patients suffering from arterial thrombosis may take benefit from curcumin [121].

In another in vitro study, eight natural products including curcumin, were compared to prednisolone regarding anti-inflammatory potential. Several pathways of inflammatory response (such as IL-6, IL-8, and TNF-α, ROS production) were investigated along with platelet activation in the blood. Besides curcumin, epigallocatechin gallate and berberine chloride also displayed good anti-inflammatory potential that suggested these compounds were alternatives to prednisolone [122].

Curcumin has been found to inhibit arachidonic acid, adrenaline, and collagen-induced aggregation of platelets. It blocked the formation of thromboxane B2 (Conversion of A2 into B2) and increased the production of 12-LOX. Furthermore, the anti-inflammatory effect of curcumin is mediated via multiple mechanisms including its impact on eicosanoids biosynthesis [121], increasing the expression of PPAR-α, IL-4, platelet/endothelial cell adhesion molecule 1, netrin G1, plasma cell endoplasmic reticulum protein 1, and delta-like 1, which have been correlated with cell adhesion and migration [107].

Curcumin reduces platelets adhesion in cerebral microcirculation mainly via endothelium modulation. PDGF-βR in phosphorylated form, extracellular signal regulated kinase (ERK-1/2) epidermal growth factor receptor (EGFR), and c-Jun N-terminal kinase (JNK1/2) levels were decreased by curcumin due to increase in the activity of PPARγ [123]. Curcumin attenuated cigarette smoke-induced elevation in AMP, ATP, and decreased ADP hydrolysis in rats. These effects of curcumin are due to modulation of purinergic signaling, platelet aggregation, and thrombus formation regulation [124].

### 4.3. Autophagy

Curcumin cause autophagy induction in platelets indicated by inhibition of PKB, and activation of AMP kinase [109]. Cellular autophagy is linked with cell death and survival [125,126]. Autophagy in platelets is observed during activation and play an important role in hemostasis and thrombosis [127,128]. Increased autophagy reduces apoptosis and increase platelets viability in immune thrombocytopenia [129], and oxidative stress in platelets of diabetic patients [130]. In one study, curcumin potentiated platelets apoptosis in a low concentration (5 µM) and inhibited apoptosis in a high concentration (50 µM). At a low concentration, the viability of platelets was unaffected, but at a high concentration it was reduced by 17%. Moreover, curcumin inhibited the activity of P-glycoprotein in platelets [39].

### 4.4. Oxidative Status of Platelets

Several studies have shown the antioxidant effects of curcumin in platelets. Oxidative stress leads to the development of several disorders including CVDs [131]. In an in vitro study, curcumin inhibited the formation of thiobarbituric acid reactive substances (TBARS) formation in platelets in peroxynitrite-induced oxidative stress model. A 50% reduction in the level TBARS was observed at 50 mg/mL concentration of curcumin in platelets. It was concluded that curcumin display antioxidant and protective effect against damage to platelets caused by ROS/RNS [132]. In humans, administration of curcumin also exhibited antioxidant potential. Signaling pathways of ROS elicit epigenetic and transcriptional dysregulation, causing activation of platelets, chronic inflammation, and endothelial dysfunction [131]. It has been observed in several studies that curcumin display antioxidant activity in platelets facing oxidative stress.

In another study conducted in humans, curcuminoids were administered (at 500 mg/day) to thalassemic patients for one year. Curcuminoid administration elevated the plasma level of some proteins and reduced their oxidative effects. Similarly, antioxidant enzymes, oxidative stress parameters, and platelet factor-3-like activity were improved. Curcuminoids have been demonstrated to inhibit cyclo-oxygenase and 12-lipoxygenase activities in human platelets, thus showing antioxidant activity [110,111]. Moreover, curcumin inhibited damage to the cells as it is powerful antioxidant and free radicals scavenger [115].

### 4.5. Platelets Count

The effect of curcumin on platelets count has been demonstrated in several studies. One study reported that profenofos-induced reduction in platelets count was attenuated by administration of curcumin (120 mg/kg) to mice for 30 days [112]. In another study, curcumin displayed anti-inflammatory effect, however no effect on platelets count was observed, so that is a controversial scenario that needs further investigations [133]. Another study also reported increase in platelets levels by curcumin [112]. In rats administered a high fat diet (HFD), the level of total cholesterol, total lipids, C-reactive protein, TNF-α, platelet count and fibrinogen contents were elevated. Administration of curcumin (20 mg/kg, for 3 months, p.o) countered all these changes. Therefore, curcumin could be a possible choice in HFD-associated CVDs; however, it needs future exploration in humans [134]. Nanocurcumin has exhibited more effective activity in preventing chemotherapy-induced thrombocytopenia in mice. Nanocurcumin administration could preserve bone marrow integrity and increase the number of circulating platelets [135].

### 4.6. Effect on Platelet Derived Growth Factor (PDGF)

Growth factors, such as PDGF, are secreted by platelets during vascular damage and play a key role in the remodeling of vessels during extracellular and cellular response to injury [136]. PDGF is responsible for the migration, proliferation, and collagen synthesis in vascular smooth muscle cells [136]. Atherosclerosis is characterized by over expression of PDGF in arteries after inflammatory-fibroproliferative response [137]. The over-expression of PDGF also occur in fibrosis of several other organs, including liver, and lungs [136]. PDGF control platelet aggregation by a feedback mechanism on vascular system. Activation of PDGF decrease the aggregation of platelets [138]. The Stat molecules family has been demonstrated to bind to the activated PDGF-βR and to be phosphorylated following PDGF stimulation. Curcumin inhibit PDGF-mediated effects of smooth muscle cells, thus ameliorating atherosclerosis and fibrosis [136].

Studies have shown that curcumin is a beneficial drug for the treatment of PDGF related diseases. For instance, curcumin has attenuated fibrotic injury and sinusoidal angiogenesis induced by carbon tetrachloride in rats by inhibition of expression of vascular endothelial growth factor (VEGF) in hepatic stellate cells (HSCs), which is mediated by disruption of the mTOR and PDGF-βR/ERK pathways. Moreover, in conclusion, HSCs targeting is mediated via activation of the PPAR-γ dependent mechanism. Therefore, pathological angiogenesis in liver fibrosis may be reduced by targeting PPAR-γ [113].

## 5. Discussion

This review focused on regulatory effects of curcumin on platelets. Curcumin (Diferuloylmethane), a bioactive compound isolated from the roots and rhizome of *Curcuma longa* is extensively used in spices in the subcontinent and for the treatment of several diseases. Curcumin belongs to the curcuminoids class of phytochemicals and has been approved as “Generally recognized as safe” by the FDA [139]. It has good tolerability and a wide margin of safety at 4–8 g/day [140]. The emergence of curcumin as an important functional food is linked with several studies that demonstrated its antioxidant potential, anti-tumor, antidiabetic, anti-atherosclerotic, and usefulness in colitis, pancreatic, and hepatic diseases [141,142,143,144]. Its effectiveness in cancer is mediated by inhibition of COX-2, MMP-9, and NF-kB [145,146].

Although curcumin exhibited pleiotropic activity on the platelet regulation, its poor solubility and low bioavailability mainly due to hepatic and intestinal glucoronidation limits its usefulness [147]. It was reported in 2004 that oral administration of curcumin (450–3600 mg per day) to human results in undetectable plasma concentration. Therefore, several techniques have been developed to enhance plasma concentration of curcumin including nano-formulations, concurrent administration with pepper, etc. [5], the details of which are beyond the scope of this review.

Curcumin caused blockade of lipoxygenase, cyclo-oxygenase, Syk kinase, followed by activation of PLCΥ and mobilization of calcium [37,119,121,148]. The inhibition of platelet activation involves stimulation of A_2A_ receptor that in turn activates protein kinase A (PKA). Another mechanism for curcumin-mediated inhibition of platelet activation involves potentiation of inhibitory effect of P2Y_12_ADP receptor inhibitor cangrelor [149].

Curcumin has been found to be an agonist of A_2A_ receptor that activates the PKA/cAMP/AC pathway in thrombocytes. Blockade of A_2A_ receptor resulted in the inhibition of PKA, which denotes that this receptor is the key player the involvement of curcumin-induced activation of PKA in thrombocytes [150].

Curcumin has been used for the treatment of many diseases either alone or in combination with other drugs. It has demonstrated anti-thrombotic activity [148] at least in part via inhibition of platelet activation. Nonetheless, procoagulant activity and pro-apoptotic potential [151,152] also inhibit platelet activation. Moreover, curcumin exposes the anionic phospholipids phosphatidyl serine (PS) on platelets surface, which indicates procoagulant and apoptotic effect in platelets [39].

## 6. Conclusions and Future Directions

In conclusion, curcumin inhibits platelet activation and aggregation and improves platelet count. Thus, curcumin is bestowed with anti-inflammatory properties; it inhibits thrombo-embolism, atherothrombosis and leukemia potential in several diseases. These diseases are major contributors of death; therefore, it is vital to understand the therapeutic impact of phytochemicals.

Curcumin exhibited anti-platelet activity in several ways including platelet activation, aggregation, and adhesion. Currently available anti-platelet drugs are based on natural products that include aspirin, snake venom-based peptides, molecules irreversibly blocking P2Y12 [39], warfarin, heparin [153], clopidogrel and abciximab [154]. Despite the development of several anti-thrombotic drugs, their effects on morbidity and mortality are not completely known [155]. In future, this scenario will be more challenging with a persistent increase in the incidence of metabolic syndrome, thromboembolism, and CVDs. The sub-optimal activity of these drugs is due to side effects (GIT dysfunction, bleeding) and drug resistance [156]. Hence, novel therapeutics are urgently needed to reduce the adverse effects of these drugs without reducing efficacy. Curcumin shows pleiotropic activities and affects the coagulation pathway through multiple mechanisms, exhibits synergistic potential, and reduces the adverse effects associated with current drugs [157,158,159]. Thus, clinical trials must be conducted to fully evaluate the untapped potential of curcumin on platelets and offer better treatment to patients suffering from thromboembolic, leukemia and cardiovascular disease.

## Figures and Tables

**Figure 1 biomedicines-10-03180-f001:**
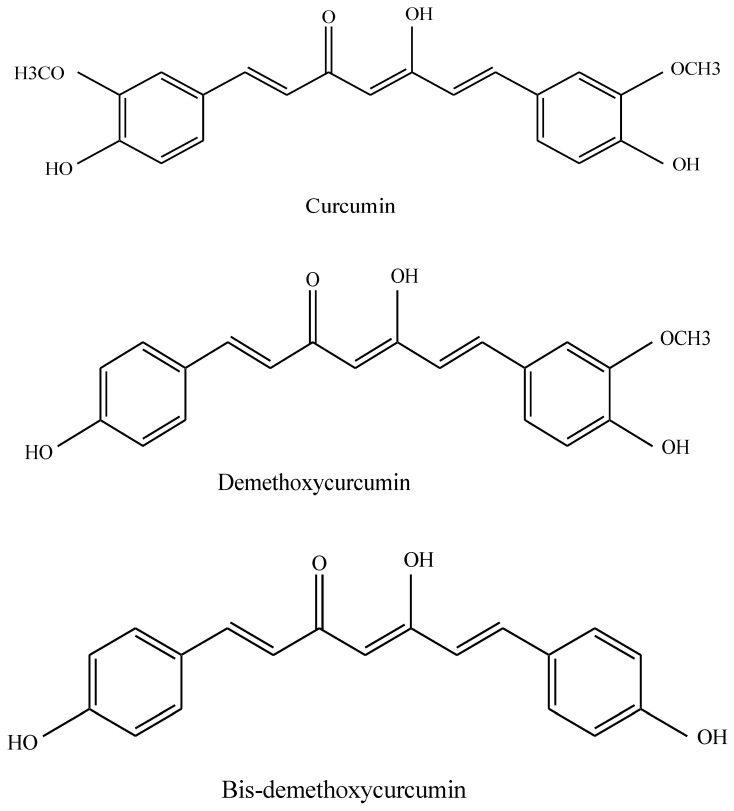
Structures of curcumin, demethoxycurcumin and bis-demethoxycurcumin.

**Figure 2 biomedicines-10-03180-f002:**
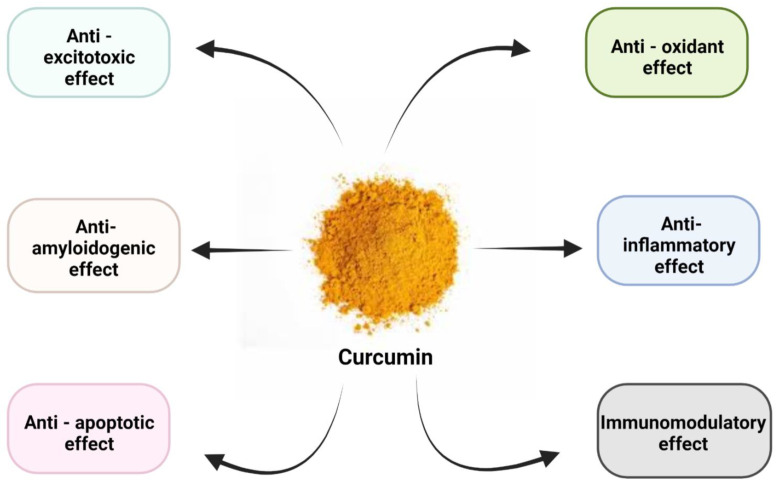
Pharmacological effects of curcumin.

**Figure 3 biomedicines-10-03180-f003:**
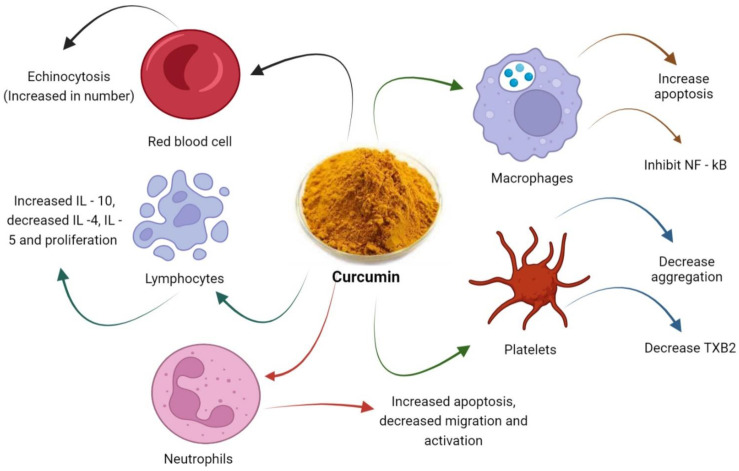
Effects of curcumin on platelets and other blood cells.

## Data Availability

Not applicable.

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
