# Peer review of "Regulatory Effects of Curcumin on Platelets: An Update and Future Directions"

_biomedicines, 2022, doi:10.3390/biomedicines10123180_

Round 1

Reviewer 1 Report

The current review article by Hussain et al reports a range of regulatory effects of curcumin in platelets. While the article is informative, it is hard to follow and requires extensive modifications to improve readability.

1.      Section 2 does not add much as the focus of the article should be on platelets and not on the general therapeutic effects of curcumin.

2.      Section 3 requires major rewriting and reshuffling. There is a lot of information, which is unclear. While reporting a study’s findings, it is important to make the information clear. Half information is hard to follow. For example,

-        The statement associated with references 60 and 61 on page 5 is hard to comprehend. Adhered to what? Reduced aggregation against what stimulation? The information related to PDGF is unclear.

-        So is the statement on page 5 – “It has been observed in several studies that curcumin improves the platelets levels”. Platelet levels under what conditions? Physiological or diseased?

-        All anticoagulant tests proved the anticoagulant activity of curcumin.

This issue is present throughout the manuscript. Section 3 is the most important section of the article. However, the subsections under it are summarized rather vaguely. More details regarding the findings of the studies that have been discussed in the review must be provided.

3.      The text under the subsections of section 3 can be further categorized into in vitro and in vivo studies.

4.      In subsection 3.5 - It is unclear whether curcumin promoted platelet production or inhibits platelet production. Does curcumin increase platelet count in case of thrombocytopenia and reduced the same in case of Obesity? This is hard to follow and requires more clarity.

5.      While each subheading has its distinct heading, they are still talking about multiple aspects related to curcumin that should be in their own subsection. There is a lot of spillages in information from one section to another. For example, Section 3.1 is on coagulation, and it also mentions platelet levels. Section 3.7 on platelet aggregation mentions the anti-inflammatory effects of curcumin on platelet adhesion and migration.

Subsection 3.2 (activation of platelet) and Subsection 3.7 (Platelet aggregation) can be combined into one.

6.      The major issue with the use of curcumin as a therapeutic agent is its reduced solubility, which significantly affects its bioavailability. Are there studies that have investigated the effects of nano-formulations of curcumin on platelet activation?

7.      Are there toxic effects of curcumin on platelets?

8.      The article would benefit from a separate section that summarizes the studies performed to investigate the effects of curcumin in different disease models.

9.      The only table described in the article is rather short. It would be ideal to summarize the important studies on platelets discussed in section 3 using a table.

10.   Figure 2 doesn’t add much to the article. It would be better to make a figure that summarizes the different mechanistic effects of curcumin on platelets along with potential molecular targets. 

Author Response

  1. Section 2 does not add much as the focus of the article should be on platelets and not on the general therapeutic effects of curcumin.

Response: A full section (Section 2) added on the functions of platelets

  1. Section 3 requires major rewriting and reshuffling. There is a lot of information, which is unclear. While reporting a study’s findings, it is important to make the information clear. Half information is hard to follow. For example,

-        The statement associated with references 60 and 61 on page 5 is hard to comprehend. Adhered to what? Reduced aggregation against what stimulation? The information related to PDGF is unclear.

-        So is the statement on page 5 – “It has been observed in several studies that curcumin improves the platelets levels”. Platelet levels under what conditions? Physiological or diseased?

-        All anticoagulant tests proved the anticoagulant activity of curcumin.

This issue is present throughout the manuscript. Section 3 is the most important section of the article. However, the subsections under it are summarized rather vaguely. More details regarding the findings of the studies that have been discussed in the review must be provided.

Response: The section has been re-visited and rectified accordingly. The sentences were re-phrased and highlighted.

  1. The text under the subsections of section 3 can be further categorized into in vitro and in vivo studies.

Response: Each subsection under section 3 contains both in-vitro and in-vivo studies in a single subsection, therefore we could not separate them.

  1. In subsection 3.5 - It is unclear whether curcumin promoted platelet production or inhibits platelet production. Does curcumin increase platelet count in case of thrombocytopenia and reduced the same in case of Obesity? This is hard to follow and requires more clarity.

Response: The subsection has been re-phrased accordingly

  1. While each subheading has its distinct heading, they are still talking about multiple aspects related to curcumin that should be in their own subsection. There is a lot of spillages in information from one section to another. For example, Section 3.1 is on coagulation, and it also mentions platelet levels. Section 3.7 on platelet aggregation mentions the anti-inflammatory effects of curcumin on platelet adhesion and migration.

Response: Platelets level statement moved from section 3.1 to 3.5. Reshuffling of statements done as highlighted in the manuscript.

Subsection 3.2 (activation of platelet) and Subsection 3.7 (Platelet aggregation) can be combined into one.

Response: Rectified as suggested

  1. The major issue with the use of curcumin as a therapeutic agent is its reduced solubility, which significantly affects its bioavailability. Are there studies that have investigated the effects of nano-formulations of curcumin on platelet activation?

Response: As curcumin has low solubility and low bioavailability. Therefore, several nanotechnological approaches have been used to overcome this issue (Reference 140, 153).  

  1. Are there toxic effects of curcumin on platelets?

Response: Yes, curcumin may cause toxic effect on platelets (bleeding).

  1. The article would benefit from a separate section that summarizes the studies performed to investigate the effects of curcumin in different disease models.

Response: Section 3 explains pharmacological effects of curcumin against different diseases.

  1. The only table described in the article is rather short. It would be ideal to summarize the important studies on platelets discussed in section 3 using a table.

 Response: Rectified accordingly

  1. Figure 2 doesn’t add much to the article. It would be better to make a figure that summarizes the different mechanistic effects of curcumin on platelets along with potential molecular targets. 

Response: Figure 2 is rectified accordingly.

Regards

Prof. Dr. Haroon Khan

Reviewer 2 Report

In this review Hussain et al. have summarized the effects of curcumin on platelet function as well as disease prevention and benefits for patients suffering from several different diseases. The review seems to be quite complete with regard to the known in vitro and in vivo effects of curcumin, and may provide valid information for further exploiting this compound. However, one should be careful with translating in vitro and animal studies to humans and patients. There are several points that I believe need to be addressed, which may help to improve the manuscript.

1.     The majority of the reviewed articles required relatively high doses (10-4 to 10-5 M) of curcumin to be active on platelets in vitro, and these doses may be even higher in the presence of plasma. This in general explains the relatively small effect sizes on platelets observed in nutritional intervention studies, in which relatively low doses are ingested. Therefore, specific dietary interventions, including curcumin, are not expected to strongly impact the prevention or treatment of diseases, purely by affecting platelet function. Although direct supplementation can have stronger effects, but the current high dosing is a strong drawback for current therapeutical interventions. This should be mentioned in the discussion, and strong claims with regard to the treatment and preventions of disease need to be minimalized (see also point 10 and 16). 

2.     As with all anti-platelet drugs, there is an increased risk of bleeding. Is there anything known about the effect of curcumin on bleeding?

3.     In the abstract the aim of the review is missing: what is the purpose? The aim is mentioned in the introduction (last sentence). Here it is stated that “the aim is to investigate the regulatory effects of curcumin on platelets”, but since this is a review no experiments have been performed, which is suggested by the word “investigate”. This needs to be rephrased. 

4.     English language needs serious checking.

5.     Introduction, Page 2: Platelets help to regulate… coagulation cascade. This sentence implies that the coagulation cascade as such is a pathological process, which is obviously not the case. This sentence needs to be rephrased, or perhaps the authors meant to say “thrombosis” instead of “coagulation cascade”? Furthermore, in the same sentence, the word “cancer” or “tumor” should be added before metastasis.

6.     Figure 1: Anti-amyloidogenic and anti-inflammatory contain a hyphen (-), while the other words beginning with anti, do not. This should be made the same for all.

7.     Page 4, paragraph 3.1: The authors state that “Coagulation is mediated by adhesion molecules present in platelets”. This is not correct, as coagulation can also occur in the absence of platelets. Platelets can promote coagulation by exposure of phosphatidylserine, which is a phospholipid and not an adhesion molecule. The reference for this sentence [53], is not complete (journal is missing), so the information stated by the authors cannot be checked.

8.     Page 4, paragraph 3.1: The sentence “Thrombin…(IL-4, IL-6, and IL-8)” is unclear. What is meant with “the induction of interleukin-1beta”? and how can IL-1b then secrete other cytokines? Please revise.

9.     Page 5, paragraph under the Table: “In in-vitro…aggregated”. The number of adhered thrombocytes to what? And why was this reduced, in the presence of curcumin? Also how does the effect of PDGF relate to curcumin? The effects of PDGF on platelets are only minor, if there are any effects at all, so the relevance of this for the present manuscript is unclear. As paragraph 3.6 also deals with PDGF and contains the same information, I propose to remove this paragraph from page 5.

10.  Page 5: the statement that “curcumin can be used against HFD-associated CVDs” is an overstatement, and only based on one study in rats [63]. This needs to be tuned down, as in humans this is not been convincingly scientifically proven, and a lot more research is needed.

11.  Page 5, paragraph 3.2: “Local prothrombin factors, such as tissue factor… platelet activation”. What is meant with “local prothrombin factors”? Prothrombin is the precursor of thrombin, cleaved by activated factor X. This sentence needs to be revised.

12.  Page 5, paragraph 3.2: “tyrosine kinase in spleen”? I assume that Spleen tyrosine kinase (Syk) is meant??

13.  Figure 2: Platelets secrete thromboxane A2 and not B2 (A2 is rapidly converted into B2, which can be measured in plasma or urine). This needs to be changed in the figure, and also in the corresponding text in paragraph 3.7.

14.  Page 6, paragraph 3.3: “Curcumin… LC3II”. Please include a reference for this statement.

15.  Page 7, paragraph 3.5: “One study… 30 days”. This sentence is not finished? I assume that the study not only reported that the compounds were administered?

16.  Page 7, paragraph 3.5: the statement that “curcumin may be used as protectant against CVDs associated with” is an overstatement, and only based on one study in rats [81]. This needs to be tuned down, as in humans this is not been convincingly scientifically proven, and a lot more research is needed. See also point 9: this part is repeated from paragraph 3.1 (page 5).

17.  Page 8, paragraph 3.7: “Growth factors… angiogenesis.” Please revise this sentence, see also point 5.

18.  References: many references are not complete; Journals and page numbers are often missing.

Author Response

REVIEWER 2

  1. The majority of the reviewed articles required relatively high doses (10-4to 10-5 M) of curcumin to be active on platelets in vitro, and these doses may be even higher in the presence of plasma. This in general explains the relatively small effect sizes on platelets observed in nutritional intervention studies, in which relatively low doses are ingested. Therefore, specific dietary interventions, including curcumin, are not expected to strongly impact the prevention or treatment of diseases, purely by affecting platelet function. Although direct supplementation can have stronger effects, but the current high dosing is a strong drawback for current therapeutical interventions. This should be mentioned in the discussion, and strong claims with regard to the treatment and preventions of disease need to be minimalized (see also point 10 and 16). 

Response: The available studies are depicted in the best possible way to express the regulatory effects of curcumin on the platelets.    

  1. As with all anti-platelet drugs, there is an increased risk of bleeding. Is there anything known about the effect of curcumin on bleeding?

Response: Yes few studies have shown the effect of curcumin against bleeding that were mentioned in the last lines of the conclusion section. Moreover, effect of curcumin against bleeding is explored in gingival bleeding as well.

  1. In the abstract the aim of the review is missing: what is the purpose? The aim is mentioned in the introduction (last sentence). Here it is stated that “the aim is to investigate the regulatory effects of curcumin on platelets”, but since this is a review no experiments have been performed, which is suggested by the word “investigate”. This needs to be rephrased.

Response: Changes were made as per reviewer suggestions   

  1. English language needs serious checking.

Response:  The manuscript was thoroughly revised for English language.

  1. Introduction, Page 2: Platelets help to regulate… coagulation cascade. This sentence implies that the coagulation cascade as such is a pathological process, which is obviously not the case. This sentence needs to be rephrased, or perhaps the authors meant to say “thrombosis” instead of “coagulation cascade”? Furthermore, in the same sentence, the word “cancer” or “tumor” should be added before metastasis.

Response: Changes were made as per reviewer suggestions   

  1. Figure 1: Anti-amyloidogenic and anti-inflammatory contain a hyphen (-), while the other words beginning with anti, do not. This should be made the same for all.

Response: Changes were made  in Figure 1 as per reviewer suggestions . Figure 2 was modified 

  1. Page 4, paragraph 3.1: The authors state that “Coagulation is mediated by adhesion molecules present in platelets”. This is not correct, as coagulation can also occur in the absence of platelets. Platelets can promote coagulation by exposure of phosphatidylserine, which is a phospholipid and not an adhesion molecule. The reference for this sentence [53], is not complete (journal is missing), so the information stated by the authors cannot be checked.

Response: Changes were made as per reviewer suggestions, the sentence was corrected, a new reference was added and highlighted within the manuscript.   

  1. Page 4, paragraph 3.1: The sentence “Thrombin…(IL-4, IL-6, and IL-8)” is unclear. What is meant with “the induction of interleukin-1beta”? and how can IL-1b then secrete other cytokines? Please revise.

Response: Changes were made as per reviewer suggestions, the sentence was rephrased and made readable, a new reference was added and highlighted within the manuscript. 

  1. Page 5, paragraph under the Table: “In in-vitro…aggregated”. The number of adhered thrombocytes to what? And why was this reduced, in the presence of curcumin? Also how does the effect of PDGF relate to curcumin? The effects of PDGF on platelets are only minor, if there are any effects at all, so the relevance of this for the present manuscript is unclear. As paragraph 3.6 also deals with PDGF and contains the same information, I propose to remove this paragraph from page 5.

Response:   The section was revised and highlighted accordingly. The sentence “The number of adhered thrombocytes” has been removed as it was not directly related to the effects of curcumin on platelets.

  1. Page 5: the statement that “curcumin can be used against HFD-associated CVDs” is an overstatement, and only based on one study in rats [63]. This needs to be tuned down, as in humans this is not been convincingly scientifically proven, and a lot more research is needed.

Response:  The sentence was fine-tuned as per reviewer suggestions.  

  1. Page 5, paragraph 3.2: “Local prothrombin factors, such as tissue factor… platelet activation”. What is meant with “local prothrombin factors”? Prothrombin is the precursor of thrombin, cleaved by activated factor X. This sentence needs to be revised.

Response:  The sentence was revised as per reviewer suggestions and was highlighted within the manuscript.

  1. Page 5, paragraph 3.2: “tyrosine kinase in spleen”? I assume that Spleen tyrosine kinase (Syk) is meant??

Response:  Yes, its actually spleen tyrosine kinase, it was revised in manuscript.

  1. Figure 2: Platelets secrete thromboxane A2 and not B2 (A2 is rapidly converted into B2, which can be measured in plasma or urine). This needs to be changed in the figure, and also in the corresponding text in paragraph 3.7.

Response:  Alright, the figure was completely revised and changes were made in the text as well.

  1. Page 6, paragraph 3.3: “Curcumin… LC3II”. Please include a reference for this statement.

Response:  The sentence was revised and reference was added.

  1. Page 7, paragraph 3.5: “One study… 30 days”. This sentence is not finished? I assume that the study not only reported that the compounds were administered?

Response:  Actually the next explanation is continuation of the previous sentence. The various dose administration and then its consecutive responses in mice. The sentence was linked in the manuscript.  

  1. Page 7, paragraph 3.5: the statement that “curcumin may be used as protectant against CVDs associated with” is an overstatement, and only based on one study in rats [81]. This needs to be tuned down, as in humans this is not been convincingly scientifically proven, and a lot more research is needed. See also point 9: this part is repeated from paragraph 3.1 (page 5).

Response:  Repetition was removed and new statement along with reference was introduced into the manuscript.     

  1. Page 8, paragraph 3.7: “Growth factors… angiogenesis.” Please revise this sentence, see also point 5.

Response:  Changes suggested by the reviewer were highlighted within manuscript.     

  1. References: many references are not complete; Journals and page numbers are often missing.

Response:  References were thoroughly revised and corrected. 

Regards

Prof. Dr. Haroon Khan

Round 2

Reviewer 1 Report

The authors have addressed all my concerns. I would therefore recommend the this article for publication.

Author Response

The authors have addressed all my concerns. I would therefore recommend the this article for publication.

Many thanks

Reviewer 2 Report

The authors have addressed all issues raised by both reviewers. The manuscript has much improved, and also Figure 2 now better summarises the effects of curcumin. There are still a few minor points that should be addressed:

1. Introduction, Page 3: Platelets help to regulate… and thrombosis cascade. The word cascade needs to be removed.

2. Title of Figure 2 should be changed to “Effect of curcumin on platelets and other blood cells”, because it now not only shows the effects of curcumin on platelets.

3. References 141-150, 154-156, 162, and 164-165 do not contain the journal. Also, not all references are correctly formatted.

After correction of these minor changes, I recommend this manuscript for publication in Biomedicines.

Author Response

  1. Introduction, Page 3: Platelets help to regulate… and thrombosis cascade. The word cascade needs to be removed.

Reply: Needful correction is done.

2. Title of Figure 2 should be changed to “Effect of curcumin on platelets and other blood cells”, because it now not only shows the effects of curcumin on platelets.

Reply: Needful correction is done.

3. References 141-150, 154-156, 162, and 164-165 do not contain the journal. Also, not all references are correctly formatted.

Reply: Needful corrections are done.

Regards